# Plasma Clearance of Coagulation Factor VIII and Extension of Its Half-Life for the Therapy of Hemophilia A: A Critical Review of the Current State of Research and Practice

**DOI:** 10.3390/ijms24108584

**Published:** 2023-05-11

**Authors:** Andrey G. Sarafanov

**Affiliations:** Center for Biologics Evaluation and Research, U.S. Food and Drug Administration, Silver Spring, MD 20993, USA; andrey.sarafanov@fda.hhs.gov

**Keywords:** hemophilia A, factor VIII, von Willebrand factor, therapeutic factor VIII, extended half-life factor VIII, plasma clearance, clearance receptors, low-density lipoprotein receptor-related protein 1, efanesoctocog alfa, activity assays discrepancy

## Abstract

Factor VIII (FVIII) is an important component of blood coagulation as its congenital deficiency results in life-threatening bleeding. Current prophylactic therapy of the disease (hemophilia A) is based on 3–4 intravenous infusions of therapeutic FVIII per week. This poses a burden on patients, demanding reduction of infusion frequency by using FVIII with extended plasma half-life (EHL). Development of these products requires understanding FVIII plasma clearance mechanisms. This paper overviews (i) an up-to-date state of the research in this field and (ii) current EHL FVIII products, including recently approved efanesoctocog alfa, for which the plasma half-life exceeds a biochemical barrier posed by von Willebrand factor, complexed with FVIII in plasma, which results in ~1 per week infusion frequency. We focus on the EHL FVIII products’ structure and function, in particular related to the known discrepancy in results of one-stage clotting (OC) and chromogenic substrate (CS) assays used to assign the products’ potency, dosing, and for clinical monitoring in plasma. We suggest a possible root cause of these assays’ discrepancy that is also pertinent to EHL factor IX variants used to treat hemophilia B. Finally, we discuss approaches in designing future EHL FVIII variants, including those to be used for hemophilia A gene therapy.

## 1. Introduction

Congenital deficiency in blood coagulation factor (F) VIII (FVIII) results in excessive bleeding. The disease (hemophilia A) is treated with infusions of therapeutic FVIII concentrates, either plasma-derived (pdFVIII) or produced with recombinant DNA technology (rFVIII). Due to the relatively short FVIII plasma half-life (~12 h), such treatment requires frequent FVIII infusions (3–4 per week in prophylaxis) that calls for developing more efficient longer-acting FVIII products, in particular with an extended plasma lifetime, commonly termed half-life (EHL). Relevant protein modifications are performed via genetic and/or chemical modification of rFVIII [1], and efficient designs require understanding the molecular mechanisms of FVIII plasma clearance.

Present work overviews (i) current knowledge in this field, focusing on the structural aspects relevant to EHL FVIII designs, and (ii) current EHL FVIII products, focusing on structural and functional aspects relevant to their efficacy. We also discuss the known issue in measurements of EHL FVIII functional activity resulting in discrepancies between assays’ results and propose the underlying mechanisms therein. As future directions, we finally discuss other EHL FVIII research designs, including those applicable to hemophilia A gene therapy, an emerging approach for disease treatment [2].

## 2. Structure and Function of FVIII

FVIII is a large heterodimeric protein (Figure 1A) circulating in plasma in complex with von Willebrand factor (VWF). Within this complex, VWF stabilizes and protects FVIII from fast clearance and facilitates FVIII delivery to coagulation sites [3]. At these sites, FVIII is site-specifically cleaved (activated) by thrombin (or FXa), resulting in dissociation of VWF (Figure 1B) [4]. Then, activated FVIII (FVIIIa) binds platelets (Figure 1C) and forms a complex there with FIXa (Figure 1D), where it serves as its cofactor in proteolytic activation of FX (Figure 1E) participating in further reactions of blood coagulation [5]. FVIIIa is inactivated mainly via dissociation of its A2 domain from the A1/A3′-C1-C2 heterodimeric fragment [6].

## 3. The Relationship of FVIII and VWF

VWF is a uniquely large multimeric plasma protein (~20,000 kDa) composed of ~50 monomers. Each monomer has a multidomain structure where the D’D3 domains provide a site for FVIII binding [7]. Though each monomer is capable of binding FVIII, studies have indicated the stoichiometry of ~1:1 for FVIII and multimeric VWF [3]. Within the complex, FVIII interacts with VWF via the C1 and C2 domains [8], and the a3 peptide of the A3 domain whose sulfated tyrosine 1680 plays a critical role for this interaction [9]. By bridging these distant sites on FVIII, VWF masks an extended portion of its light chain (LCh), indicating its importance for interactions with FVIII clearance receptors.

Even though FVIII binds VWF with high affinity (K_D_ ~0.5 nM), a small fraction of free FVIII (3–5%) always exists due to its dynamic equilibrium with the complex [8]. This FVIII fraction is subjected to fast clearance with the half-life of ~2 h [10] that engages ~24% of the infused FVIII upon its binding to endogenous VWF and gradual dissociation from it. The fraction of FVIII bound to VWF is cleared via VWF clearance mechanisms with the plasma half-life of ~20 h. With contributions of both pathways, the overall FVIII plasma half-life is ~12 h [11]. Plasma levels of FVIII and VWF strongly correlate with an estimate that each 1% change in VWF level results in a respective change of FVIII level by ~0.5% [11,12]. Individual VWF levels are highly variable (in the range of 50–200%) [13].

Thus, there are two pathways of FVIII clearance: when FVIII directly interacts with its clearance receptors and when FVIII is cleared via VWF with much longer plasma half-life. By this, the half-life of VWF represents a biochemical barrier for half-life extension for any engineered FVIII if it is able to form a complex with endogenous VWF. Below, we overview receptors involved in both pathways of FVIII plasma clearance with a focus on relevant molecular mechanisms to be considered for designing therapeutic EHL FVIII.

## 4. Plasma Clearance Receptors of FVIII and VWF

Clearance receptors for FVIII and VWF (Table 1) can be classified based on the chemical nature of the ligand recognition determinants, which can be protein- or carbohydrate-based. In general, protein recognition provides more specificity in ligand binding, whereas glycan-based recognition would result in binding a broader range of ligands.

### 4.1. Low-Density Lipoprotein Receptor-Related Protein 1 (LRP1)

LRP1 was the first found FVIII clearance receptor, described independently by research groups of Saenko and Lenting in 1999 [15,28]. LRP1 is the major clearance receptor for both FVIII and VWF, based on many in vitro, animal, and genetic studies [14]. In circulation, LRP1 is expressed on hepatic cells where it internalizes ~15 ligands, such as alpha-2 macroglobulin, proteases, coagulation factors, and lipoproteins, as well as a large variety of ligands in other tissues [29,30]. LRP1, similar to other related receptors (see below), interacts with its ligands via a flexible string of adjacent complement-type repeat domains (CRs), and lysine residues on ligands are the major determinants for the interactions [31].

On FVIII, such lysine residues are located on the LCh throughout its A3, C1, and C2 domains [28,32,33]. The FVIII-LRP1 interaction (K_D_ ~20 nM) occurs via formation of multiple alternative binding combinations in a dynamic mode where a real-time interactive site involves a string with three or more CR domains of the receptor and multiple lysines of FVIII, as found by us recently [33]. Our study also showed a major role for the binding of the C1 domain, based on blocking FVIII-LRP1 interaction by an anti-C1 domain single-chain variable antibody fragment (scFv) iKM33. Both VWF- and LRP1-binding sites on FVIII LCh are extended and significantly overlap (Figure 1A).

Upon FVIII activation, another LRP1-binding site is exposed on the A2 domain. The functional relevance of that may reflect a participation of LRP1 in plasma clearance of inactivated FVIIIa remnants (A2 domain and A1/A3′-C1-C2 heterodimer). Indeed, each of these fragments is cleared from mouse plasma with the involvement of LRP1 [34,35].

On VWF, the LRP1-binding site is located within its A1 domain [36]; thus, all VWF monomers carry such sites. In contrast to VWF sites for binding FVIII, its LRP1-binding sites are not exposed under static fluid conditions in vitro due to folded state of molecule and become exposed only when it unfolds under fluid shear force in blood vessels [37].

### 4.2. Low-Density Lipoprotein Receptor (LDLR)

LDLR was identified as the second FVIII plasma clearance receptor in 2005 [16]. LDLR belongs to a large family of endocytic receptors named after LDLR itself and involving seven members in mammals, including LRP1 [38]. The family members are expressed in many tissues where they interact with numerous ligands and regulate many processes, including those with medical importance [29,30,39,40].

Expressed in the liver, LDLR catabolizes a smaller number of ligands, mainly low-density lipoprotein (LDL), relevant to cholesterol metabolism [41]. The role of LDLR in FVIII clearance was supported in a mouse model, where LDLR deficiency resulted in ~1.5-fold prolongation of FVIII half-life, similarly to LRP1. In turn, combined deficiency in both LRP1 and LDLR resulted in ~5-fold prolongation of FVIII plasma half-life with an increase in its plasma level. The authors concluded that LRP1 and LDLR cooperate in FVIII plasma clearance [16], which was supported in further genetic studies [42,43].

Compared to LRP1, the affinity of LDLR for FVIII is lower (K_D_ 30–60 nM), which may explain its apparently smaller contribution to FVIII plasma clearance. As we found previously, the organization of FVIII binding sites for both receptors is similar as they are located on the FVIII LCh with the same significance of the C1 domain for LDLR binding [44] and due to presence of the binding site also for LDLR in the A2 domain in FVIIIa [45,46]. Reanalysis of these data [44], considering the FVIII-LRP1 interaction mode [33], indicates that the mechanism of FVIII-LDLR interaction is also based on formation of alternative binding combinations.

In addition to LDLR and LRP1, other members of the LDLR family exposed to circulation are very low-density lipoprotein receptor (vLDLR) and megalin, expressed on endothelial and kidney cells, respectively. Both can interact with FVIII and FVIIIa in vitro similarly to LRP1 and LDLR, however, with unknown functional relevance [45,46]. In particular, the role of vLDLR in FVIII clearance was not supported in the mouse model [47]. We suggest that vLDLR and megalin may play a role in the clearance of inactivated FVIIIa (fragments).

### 4.3. Cell-Surface Heparan-Sulfate Proteoglycans (HSPGs)

HSPGs have covalent attachment of several negatively charged sulfated polysaccharide chains and are localized on cell membranes. There is a large variety of HSPGs which interact with numerous protein ligands via their basic residues [48,49]. HSPGs serve as receptors or coreceptors, in particular of LRP1, facilitating its ligand recognition. In plasma, such ligands involve lipoprotein lipase [50], apo E-containing lipoproteins [51,52], thrombospondin [53], thrombin-protease nexin 1 complex [54], tissue factor pathway inhibitor [55], thrombin, antithrombin III, and FX [49], and FVIII [17].

In these processes, HSPGs preconcentrate the ligands to facilitate their further interactions with other receptors. In particular, FVIII has extremely low plasma concentration (≤1 nM) [56], insufficient for its effective interaction with LRP1 (K_D_ ~20 nM). The involvement of HSPGs, abundantly expressed on cells [17], altogether with the cell membrane interaction of FVIII [57], is believed to increase its local concentration and facilitate LRP1 binding to LRP1. Similarly, HSPGs may facilitate FVIII interactions with other clearance receptors.

### 4.4. Asialoglycoprotein Receptor (ASGPR)

Other clearance receptors of FVIII and VWF (lectins) recognize their carbohydrates. Activated form of FVIII contains four N-linked glycans, whereas the dissociated B-domain contains ~14 N- and ~7 O-linked glycans [4,58]. VWF contains 23 glycans (N- or O-linked) per monomer [14,59], with an estimated number of >1000 on the molecule. In both proteins, the N-glycans contain bi-, tri-, and tetra-antennary structures with high mannose portions, ABO(H)-determinants; and the O-glycans contain mainly T-antigen. These glycans (as well as in other proteins) are terminally capped by sialic acid sugars with ≥80% occupancy [58,59,60,61,62,63]. Notably, the N-glycosylation of rFVIII differs from pdFVIII mainly in sialic acids, ABO(H) blood group, etc. The greatest difference was found in rFVIII expressed in human embryonic kidney (HEK) cells, whereas the use of baby hamster kidney (BHK) and Chinese hamster ovary (CHO) cells also resulted in such a difference [58]. The half-life of VWF is dependent on its ABO(H) group glycans, where the O-type glycan-bearing VWF variants have significantly shorter plasma half-life, associated with FVIII level as noted above [11].

ASGPR (Ashwell receptor), a member of the C-type family of lectins recognizing terminal asialic sugars [64], was found to be a clearance receptor for both FVIII and VWF. ASGPR binds with high affinity to FVIII asialylated N-glycans (K_D_ ~2 nM), where the major binding determinant is the abundantly glycosylated B-domain. However, removal of the B-domain does not significantly affect FVIII plasma half-time in various species including humans. Therefore, the authors concluded that ASGPR likely plays a role only in quality control of FVIII biosynthesis by eliminating incompletely glycosylated protein [18]. Other causes of proteins desialylation are proteins “aging” [65] and infection with pathogens [19,66]; both processes likely engage ASGPR in the clearance of such proteins.

Regarding VWF, a deficiency in ASGPR resulted in ~1.5-fold prolongation of its plasma half-life in a mouse model; in parallel, FVIII level was increased [19]. Other plasma ligands of ASGPR include chylomicron remnants, fibronectin, lactoferrin, immunoglobulin A, members of the prolactin/growth hormone family, lipoprotein(a), urokinase-type plasminogen activator [18], and platelets [19].

### 4.5. Sialic Acid Binding Immuno Globulin-like Lectin Member 5 (SIGLEC5)

In humans, there are 14 members of SIGLECs. These receptors are expressed on cells of hematopoietic origin, whereas each member is found on specific cell type(s). SIGLEC5 recognizes the two most common sialic acid sugars, N-acetylneuraminic acid and N-glycolylneuraminic acid, and is expressed on macrophages. SIGLEC5 binds to both FVIII and VWF with high affinity (K_D_s ~8 and ~14 nM, respectively). In cell culture, SIGLEC5 mediated internalization of both proteins; and in mice, overexpression of SIGLEC5 resulted in significant decrease in FVIII and VWF levels [20].

Within the FVIII/VWF complex, FVIII does not interact with SIGLEC5. In contrast to VWF, desialylation of FVIII does not affect its binding to SIGLEC5, indicating that FVIII interacts with receptors via protein moiety. Indeed, each of full-length (FL) FVIII and a B-domain deleted (BDD) FVIII, lacking ~85% of FL-FVIII glycans, had similar binding to SIGLEC5. The authors concluded that SIGLEC5 may regulate plasma level of the FVIII/VWF complex [20].

### 4.6. C-Type Lectin Domain Family 4 Member M (CLEC4M)

CLEC4M is a mannose-specific endocytic receptor expressed on endothelium of liver, lymph nodes, and placenta. CLEC4M interacts with both FVIII and VWF via their N-glycans and internalizes both proteins in cell culture; in the case of FVIII, the interaction occurred in either a VWF-dependent or independent manner. Overexpression of CLEC4M in mice resulted in decreased plasma levels of both FVIII and VWF. The authors suggested that CLEC4M is a clearance receptor of both FVIII and VWF [21,22].

This role of CLEC4M was further supported by finding association of polymorphism in the receptor gene with FVIII and VWF plasma levels (see below). Other known ligands of CLEC4M include pathogens, such as *Mycobacterium tuberculosis*, HIV-1, influenza A, SARS-CoV, Ebola, hepatitis C, and West Nile viruses [67], whereas no endogenous glycoprotein ligands besides FVIII and VWF were reported.

### 4.7. Stabilin-2 (STAB2)

STAB2 is a class H endocytic receptor, recognizing glycosaminoglycans (mucopolysaccharides), and is expressed on endothelial cells in liver, spleen, and lymph nodes. Studies demonstrated association between STAB2 genetic polymorphism and plasma levels of both FVIII and VWF [68,69,70,71]. STAB2 binds and internalizes VWF, and binds FVIII in a VWF-independent mode. In STAB2-deficient mice, the half-lives of VWF and FVIII were prolonged ~1.5-fold upon injection of VWF, whereas FVIII itself interacted weakly with STAB2. The authors concluded that STAB2 is a clearance receptor of the VWF and FVIII/VWF complex. Other ligands of STAB2 include hyaluronic acid, heparins, chondroitin sulfates, collagen, and advanced glycation end products [72].

### 4.8. Scavenger Receptor Type A Member 5 (SCARA5)

SCARA5 (SR-A5) is expressed on endothelial cells in the spleen and kidneys, where it recognizes lipopolysaccharides, ferritin, polyanions, and bacteria [24], indicating that recognized determinants are carbohydrates. SCARA5 binds VWF with very high affinity (K_D_ ~0.5 nM), whereas it does not bind FVIII. SCARA5-expressing human cells bound and internalized VWF, but not FVIII. Under the condition of SCARA5 deficiency in mice, the half-life of injected VWF was prolonged by ≤1.3 times. The authors concluded that SCARA5 is an endocytic receptor for VWF [24]. This was supported by the finding of association of SCARA5 gene polymorphism with VWF levels [69].

### 4.9. Scavenger Receptor Type A Member 1 (SCARA1)

SCARA1 (SR-A1) is localized on macrophages where it recognizes polysaccharides, lipopolysaccharides, spectrin (a marker of apoptotic cells), bacteria, extracellular matrix, proteoglycans, etc. [73], most likely via carbohydrates. SCARA1 binds to VWF with high affinity (K_D_ ~14 nM) via multiple sites suggested to be on the D’D3, A1, and D4 domains via both protein and glycan determinants. VWF bound to SCARA1-expressing cells and had slower plasma clearance in SCARA1-defcient mice. Two VWF mutants, known for increased plasma clearance in humans, had significantly higher binding to SCARA5 in a purified system and on macrophages, and slower plasma clearance in SCARA1-defcient mice. The authors suggested that SCARA1 contributes to increased plasma clearance of certain VWF variants [25].

### 4.10. Macrophage Galactose-Type Lectin (MGL)

MGL is a C-type lectin, expressed on macrophages and recognizing terminal N-acetyl galactosamine or galactose sugars. FVIII binds to MGL via asialic O-linked glycans on the B-domain. In MGL-deficient mice, FVIII level was ~1.3-fold elevated, consistent with a role of MGL in clearance of both FVIII and VWF. In VWF-deficient mice, FVIII half-life was reduced ~8-fold, but prolonged 3–4-fold upon (i) inhibition of MGL with antibodies and (ii) depletion of macrophages by a specific reagent (clodronate). The authors concluded that MGL plays an important role in macrophage-mediated plasma clearance of both FVIII and the FVIII/VWF complex [26].

In an earlier study, dose-dependent binding of VWF to MGL was observed, as well as an enhanced clearance of hypo sialylated VWF in VWF/ASGPR-deficient mice, inhibited by coadministration of an MGL antibody. In MGL-deficient mice, VWF levels were significantly elevated [27]. Altogether, these studies demonstrated a role of MGL in plasma clearance of FVIII via both VWF-dependent and independent pathways.

### 4.11. Other Factors Interacting with FVIII and VWF, or Affecting Their Plasma Levels

In addition to receptors mentioned above, other factors were also indicated to participate in FVIII plasma clearance based on less-in-depth studies. Two lectins, Galectin-1 and -3 (Gal-1, Gal-3), expressed on endothelial cells and recognizing *β*-galactoside residues, bound FVIII with high affinity (K_D_s 0.1–0.5 nM) via N- and O-linked glycans, mostly located on the B-domain. Notably, a BHK-cell-derived rFVIII had enhanced affinity to both receptors compared to the CHO-cell-derived rFVIII [74], consistent with their difference in N-glycans mentioned above [58]. VWF also bound both galectins, mostly via N-glycans but with lower affinity (K_D_s 20–80 nM) [75]. The authors concluded that Gal-1, Gal-3 may influence FVIII procoagulant activity and VWF-mediated thrombus formation [74,75].

In several studies involving more than 50,000 subjects total, genetic polymorphism (a single nucleotide or amino acid change) in many genes was associated with FVIII and VWF plasma levels. The FVIII-linked genes were *STAB2*, *VWF*, *F8*, *ABO*, *SCARA5*, *STXBP5*, *LBH*, *FAM46A*, *VAV2*, *ACCN1*, *KATNB1*, *LDLR*, and *LRP1*; and the VWF-linked genes were *STXBP5*, *SCARA5*, *ABO*, *STAB2*, *VWF*, *STX2*, *TC2N*, *CLEC4M*, *VPS8, EBP41L4A*, *KRT18P24*, and *SAFB2* [42,43,68,69,76]. A later study of Sabater-Lleal et al. identified additional 7 FVIII- and 11 VWF-linked genes/gene groups, supported by functional validation of results using cell culture where silencing of respective genes resulted in change of FVIII and VWF levels [77].

Recently, Swystun and Lillicrap described particular variants of *LDLR*, *ASGPR*, *CLEC4M*, *TC2N*, and *ABO(H)* affecting the plasma half-lives of FVIII and VWF in support of development of personalized treatment plans for patients with hemophilia A [11]. It should also be noted that elevated FVIII levels are associated with increased risk of thrombosis [78]. Overall, the genetic studies supported the role of FVIII and VWF clearance receptors reviewed above, whereas the roles of other identified genes remain to be explored.

### 4.12. Summary—FVIII Determinants to Be Modified to Extend the Plasma Half-Life

Knowledge derived from understanding FVIII clearance mechanisms indicates that major FVIII determinants to clearance receptors are located on the LCh. This portion of FVIII carries extended binding sites for LRP1, LDLR, and VWF, for which interactions with FVIII significantly affect its plasma half-life. Therefore, blocking these sites on FVIII may potentially decrease or abolish these interactions and extend FVIII plasma half-life.

## 5. Designing FVIII Molecule to Extend Its Plasma Half-Life

### 5.1. Approaches to Modify FVIII

Relevant modifications of FVIII are based on two approaches. The first approach is to affect the pathway where FVIII directly interacts with clearance receptors. If the modified FVIII retains the ability to interact with endogenous VWF, the half-life extension would be limited by that of VWF (~20 h). The second approach is to affect both pathways of FVIII plasma clearance (Section 3) with additional disrupting of the FVIII interaction with VWF to extend FVIII plasma half-life beyond that of VWF.

The potential and used FVIII designs involve (i) shielding its LRP1/LDLR-binding sites on the LCh, together with (ii) shielding its VWF-binding site (to decouple FVIII from the VWF-dependent plasma clearance pathway), (iii) increasing FVIII affinity to VWF, (iv) affecting FVIII carbohydrate moieties, and (v) making FVIII single-chained (SCh) which stabilizes the molecule by preventing its dissociation during purification, etc. Notably, the SCh form retains FVIII major functions [79] and was found to be present in all preparations of rFVIII [80]. In many designs, (vi) removal of the large, heavily glycosylated B-domain, believed dispensable for FVIII activity [81], is considered favorable for both the manufacturing process and gene therapy applications.

Other FVIII modifications use common approaches approbated on many biologicals. These are (i) covalent attachment of polyethylene glycol (PEG), increasing the molecule’s hydrodynamic radius, which interferes with plasma clearance [82], (ii) genetic fusion with an antibody Fc-fragment resulting in protein retention/recycling by vasculature, also prolonging its half-life [83], and (iii) genetic fusion with a hydrophilic unstructured low-immunogenicity polypeptide(s) (XTEN) acting similarly to the PEG moiety [84]. The majority of the above modifications have been used to design current therapeutic EHL FVIII products, which are reviewed below with a focus on the structure-function aspects.

While it is commonly assumed, it should be noted that the above modifications are performed using the rFVIII platform. This allows performing protein modifications at the genetic level, while pegylation is performed on purified protein. Notably, all rFVIII products contain a protein unable to bind endogenous VWF, in contrast to pdFVIII. This fraction may constitute ≤20% of total protein and, having fast plasma clearance rate, may affect its overall plasma half-life upon infusion [80].

### 5.2. Discrepancy in Activity Measurements of Modified FVIII Variants

Two types of assays are used to determine FVIII activity for assigning the potency of FVIII products and monitoring FVIII plasma levels in clinics [85]. The first is the one-stage clotting assay (OC), in which plasma deficient in FVIII is spiked with test FVIII, and the clotting time is measured versus that of samples with FVIII standard with known activity (in international units, IU). The assay principle is based on modeling the intrinsic pathway of blood coagulation.

The second is a chromogenic substrate (CS) assay, which is based on testing the FVIII-dependent step of the coagulation pathway. In this assay, the test FVIII is diluted in FVIII-deficient plasma, treated with thrombin to activate FVIII, and then added to a mix of FIXa and FX (and phospholipid), allowing FVIIIa to act as cofactor of FX to generate FXa (step 1). When the reaction is complete, a chromogenic substrate of FXa is added and the reaction product is measured (step 2). This amount is proportional to the amount of generated FVIIIa, whose activity (IU) is extrapolated from comparison with FVIII standard samples.

Both OC and CS assays are available in multiple reagent combinations, which may report different results of activity (IU) for the same FVIII sample and FVIII standard. In most cases, the assay type or reagent-dependent difference in FVIII activities is not random or erroneous but caused by a bias between the assays based on different reagents. Such bias may be indicative of the analytical difference of the test FVIII sample and FVIII standard, for example, when FVIII sample is a purified recombinant protein and the FVIII standard is pooled plasma, or when the standard is a plasma-derived protein. In these measurements, the OC assay has the highest discrepancy in results.

Another source of the assay discrepancy is the nature of FVIII modification(s). For the wild-type full-length (FL) rFVIII, tested versus purified pdFVIII standard, the OC and CS assays generally produce similar results, whereas these may be discrepant for a modified FVIII. In particular, deletion of the B-domain may result in lower value of FVIII activity measured by OC assay up to ~0.5-fold of the CS assay result [86,87]. The OC/CS assay discrepancy causes an uncertainty in FVIII potency assignment; thus, selection of appropriate method for that should be supported by a clinical study. Due to the importance of these matters for efficacy of EHL FVIII products, we also overview the activity measurements for each FVIII variant below.

## 6. Extended Plasma Half-Life FVIII Variants

Current EHL FVIII products available in the US are shown in Table 2. 

### 6.1. Efmoroctocog Alfa

This EHL FVIII variant (also known as rFVIII-Fc) is based on the deletion of the B-domain and genetic fusion with a human immunoglobulin Fc-domain via the FVIII C-terminus. Thereafter, we use a commonly accepted term “B-domain deleted” (BDD); however, it corresponds to deletion of the major portion of the B-domain while preserving several amino acids at its each flank to keep integrity of the respective thrombin and furin cleavage sites. The rFVIII-Fc preparations may contain up to 39% of the SCh form. Compared to an FL-rFVIII competitor [93], rFVIII-Fc showed ~1.5-fold longer plasma half-life in clinical studies. The potency is assigned using a CS assay with recommended administrations once every 4 days [88].

Upon rFVIII-Fc thrombin cleavage (activation), the molecule retains the Fc portion [94]. It is of interest if the Fc fragment, attached close to the protein contact point with platelet membrane (Figure 1C), affects this and other interactions of activated protein within the tenase complex and, respectively, the rate of FX conversion into FXa, although such potential effect is likely small since rFVIII-Fc therapeutic dosing is comparable to other FVIII products. Yet, in vitro data [94] indicate the possibility of slower FX conversion. In addition, the presence of considerable amount of the SCh [33] may result in overall slower activation of rFVIII-Fc as it requires cleavage at its two sites to release endogenously bound VWF that may contribute to the assay’s discrepancy.

Early studies reported the OC/CS assay ratio as ~0.9 for rFVIII-Fc activity measurements [94,95], whereas in subsequent studies, it was as low as ~0.5 for some reagents combinations and some patients’ plasma samples [96,97,98]. Similar OC/CS ratios were found for other BDD-FVIII-based variants: turoctocog alfa, simoctog alpha, and moroctocog alfa [87,97]. This indicates that all BDD-FVIII variants may show such discrepancy, though particular assay setups may minimize it.

### 6.2. Rurioctocog Alfa Pegol

This EHL FVIII variant (BAX 855) is based on an FL-FVIII, octocog alfa [93]. BAX 855 contains ~2 molecules of PEG (MW ~20 kDa), covalently attached to lysines by a chemical process resulting in the random linking reported to be mostly (~60%) within the B-domain [99]. Compared to parental FL-FVIII, BAX 855 demonstrated ~50% reduced binding to LRP1 [99], and ~1.4-fold longer plasma half-life in clinical studies [100,101]. The potency is assigned by an OC assay, while a field study with patient plasma samples showed lower OC results (<20%) than those in a CS assay [102,103]. Recommended frequency of administrations is twice a week [89].

A relevant question is whether the random pegylation of FVIII results in the appearance of a fraction of nonfunctional FVIII molecules. Indeed, FVIII has a large amount of surface-exposed lysines (PDB ID 6MF2), including those located within its sites for interactions with respective plasma ligands and capable in PEG attachment. In any case, such protein fraction is likely neutral in physiological conditions.

### 6.3. Damoctocog Alfa Pegol

This EHL FVIII variant (BAY 94-9027) is based on a BDD-FVIII with a chemical attachment of a double-branched PEG molecule (~60 kDa) to cysteine 1804 introduced via the point-mutation on the A3 domain [104]. The protein plasma half-life was extended by ~1.5 fold compared to an FL-FVIII in clinical studies [105]. The potency is assigned by a CS assay with recommended administrations twice a week [90].

The OC/CS assays relationship for measuring BAY 94-9027 activity in plasma was investigated in a field study involved 52 clinical laboratories, which analyzed plasma samples spiked with BAY 94-9027 and a comparator, FL-rFVIII (octocog alfa), using their routine OC and CS assays, and OC assay protocols (kits) provided by the BAY 94-9027 manufacturer. An analysis of data indicates that the in-house OC assays produced ~25% lower activity than CS assay (Appendix A). At the same time, the OC assay reagents provided by the BAY 94-9027 manufacturer resulted in better protein recovery and precision [106].

It is interesting to know why pegylation of the BDD-FVIII at position 1804 does not interfere with the protein interactions within the tenase complex, as this position is located close to the region 1811–1818 comprising the binding site for FIXa [107]. Indeed, the attached nearby PEG has a large hydrodynamic size due to high hydrophilicity.

### 6.4. Turoctog Alfa Pegol

This EHL FVIII variant (N8-GP) is based on a BDD-FVIII with enzymatic linkage of a PEG (~40 kDa) to an O-glycan within the B-domain linker. Upon protein activation, the PEG moiety and the linker are altogether cleaved off, and the resulting FVIIIa acquires structure similar to pdFVIIIa. N8-GP has ~3-fold reduced binding to LRP1 [108], and, in clinical studies, had plasma half-life of 17–22 h (~1.6-fold extension) compared to parental BDD-FVIII [109]. The potency is assigned by a CS assay with recommended administrations once every 4 days [91].

A study of the N8-GP manufacturer showed that a stepwise increase of PEG size from 10 kDa to 80 kDa did not significantly affect FVIII activity measured by a CS assay. In turn, an OC assay produced similar activity (IU) with the use of one OC assay reagent, while it resulted in lower values with other reagents (contact pathway activators, etc.) [108]. The OC/CS assays relationship for measuring N8-GP activity in plasma was further investigated in a field study involving 67 clinical laboratories. Participants used their routine methodologies for analysis of plasma samples spiked with N8-GP and a comparator, FL-rFVIII [110]. Analysis of these data indicates that OC assay generally produced N8-GP activity lower by ~25% versus CS assay (Appendix A). In another study, an OC assay also produced lower N8-GP activity (by ~30%) in plasma from 21 patients treated with N8-GP [111]. However, using selected OC assay reagents by the N8-GP manufacturer resulted in activity values closer to those by CS assay [112].

### 6.5. Efanesoctocog Alfa

This EHL FVIII variant (BIVV001) [113] combines modifications of several types: B-domain deletion, fusion of the HCh and LCh into single-chain, and C-terminal Fc-fragment fusion (with one of two chains of the fragment). The new modifications are (i) attachment of the VWF FVIII-binding D’D3 domains fused via (ii) thrombin-cleavable linker to the second chain of Fc-fragment, and (iii) insertion of two XTEN polypeptides in place of the B-domain (288-amino acid (aa) residues) and between the D’D3 fragment and the linker (144-aa residues). Upon thrombin activation, the D’D3 and both XTEN fragments, as well as the N-terminal LCh peptide, are cleaved off, and the resulting FVIIIa (retaining the Fc-fragment) is similar to activated efmoroctocog alfa [114].

The key feature of BIVV001, in contrast to the other EHL FVIII products, is that it does not bind endogenous VWF, making BIVV001 independent of the VWF clearance pathway. Consistently, in clinical studies, BIVV001 demonstrated 3–4-fold longer plasma half-life relative to standard FVIII products [115]. The potency is assigned using an OC assay, while a CS assay overestimates BIVV001 activity by ~2.5-fold [92] corresponding to the OC/CS ratio of ~0.4. Recommended frequency of administrations is once a week.

Thus, BIVV001 has the largest discrepancy in the OC/CS assay ratio compared to other EHL variants. We suggest that this can be due to (i) the slowest activation of BIVV001 because of the requirement of three specific cleavages to release the D’D3 fragment, (ii) deletion of the B-domain, and (iii) retention of the Fc-fragment by activated protein, as discussed above for rFVIII-Fc, to which both factors the OC assay may be sensitive. Notably, the decoupling of BIVV001 from endogenous VWF seems not affected its clinical efficacy [115].

### 6.6. Lonoctocog Alfa

This FVIII product (rFVIII-SC) is not considered an EHL FVIII; however, we include its description here because (i) its design is relevant to the BIVV001 variant of EHL FVIII, and (ii) the activity measurements results with rFVIII-SC are relevant to our discussion in Section 7.2. (below). rFVIII-SC is based on a BDD-FVIII with deletion of four amino acid residues by the furin cleavage site to make the molecule uncleavable. The latter results in formation of the single-chain protein with moderately higher affinity to VWF than an FL-FVIII [116]. Upon thrombin cleavage (activation), rFVIII-SC acquires a structure similar to pdFVIIIa [79]. The potency is assigned using a CS assay with recommended administrations 2–3 times a week [117].

The relationship between OC and CS assays for measuring FVIII-SC activity in plasma was investigated in a field study involved 28 clinical laboratories that found that the protein exhibits ~50% lower activity by OC assay [118]. By considerations in Section 6.1., we suggest that the underlying cause of this assay discrepancy is slower activation rate of rFVIII-SC due to the single-chain structure of this protein requiring two activation cleavages to dissociate the complexed endogenous VWF.

### 6.7. Summary—EHL FVIII Drug Products

Current EHL FVIII products have significant FVIII modifications which are based on change of protein moiety and attachment of other moiety of protein or nonprotein nature. Protein moiety alterations are performed via gene engineering, and the nonprotein moiety (PEG) is attached to purified protein chemically or enzymatically. The majority of these modifications affect the pathway of free FVIII clearance, while modifications in the most recent product, BIVV001, also affect the VWF-mediated pathway, breaking the biochemical barrier posed by the VWF half-life and resulting in significant extension of the plasma half-life. Typically, these modifications affect the protein activity measurement in OC assay, resulting in lower values compared to CS assay values. In all cases, the appropriate product dosing was justified in clinical studies.

## 7. Discussion

### 7.1. The OC/CS Assay Discrepancy in the Measurements of EHL FVIII Variants Activity

Measurements of EHL FVIII activity in plasma generally produce the OC assay activity values 20–60% lower compared to the CS assay values. Among factors contributing to the assays’ discrepancy are differences in analytical conditions (reagents, etc.), and structural difference of the EHL variants relative to FL-FVIII-based standards. Here, we aimed to assess the latter component for two structurally similar EHL variants, pegylated and BDD-based proteins, using an analysis of published data.

In the respective field studies with BAY 94-9027 and N8-GP (Refs. [106,110] and Appendix A), the OC assay recovery was much better (~100%) for the lowest amount of spiked protein (0.03 IU/mL) while it progressively decreased to the highest spiked amount (0.9 IU/mL; ~80% recovery). In the CS assay, the trend was the opposite, with overestimation of protein recovery at each concentration. The resulting OC/CS assay ratio was the highest to lowest for the lowest to highest spiked protein amounts, respectively. Similar trends were found for respective FL-FVIII comparators, while their recovery values were shifted relative to the values of the EHL FVIII, being higher in the OC assay and lower in the CS assay. In Appendix A, we tabulate and plot these data on graphs.

To assess the assay discrepancy components related to the structural differences of each EHL variant and the FL-FVIII comparator, we determined their differences in the OC/CS values for each spiked protein concentration. It resulted in strikingly similar differences of the OC values from the respective CS values, ranging from 0.33 to 0.2–0.25 from the lowest to the highest spiked amount of protein and corresponding to the average OC/CS ratio of ~0.75 for each BAY 94-9027 or N8-GP (Appendix A). We attribute such differences to both B-domain deletion and PEG addition in each EHL variant. Again, these differences were almost identical for both EHL variants, which in turn have general structural architecture with differences in size of attached PEG, its attachment site on FVIII and in the B-domain linker. Such an approach can be likely applied to other modified FVIII variants to assess the effect of particular modification(s) on the OC/CS assay results discrepancy.

### 7.2. Possible Root Cause of the OC/CS Assay Discrepancy for EHL FVIII/FIX Variants

Various factors resulting in OC/CS assay discrepancy are described in the literature. Among the factors related to protein structure, these are naturally occurring mutations in FVIII affecting its activation rate, stability of FVIIIa, and its interactions with components of tenase complex, causing mild or moderate hemophilia. In the EHL variants, the PEG moiety was suggested to interact with some components of the OC assay (contact pathway activators), resulting in prolongation of the clotting time and, respectively, lower values of FVIII activity compared to CS assay [85]. However, the root cause of the lower results in the OC assay remains poorly understood.

To explain the underlying factors of this discrepancy, we suggest the following. We assume that OC assay is sensitive to the overall *dynamics* of the clotting reactions, depending on rate of FVIII activation and rate of consequent FXa generation. Due to this, the slower thrombin cleavage (activation) rate of an EHL-FVIII compared to FL-FVIII (a comparator) postpones the clotting time, and, consequently, is determined with less activity unitage. In contrast, a CS assay, being a two-step, by this rather *static* procedure is likely less sensitive to the thrombin cleavage rate of FVIII, as it measures total cleavable (activatable) protein; thus, it would produce higher activity unitage as subsequent reactions cannot occur earlier due to their physical separation.

This implies that at the initial phase of activation of a slower activatable FVIII in OC assay, small amount of it triggers the clotting pathway, while another fraction of the protein at the late phase of its activation may contribute less to timing of the clotting and not even be fully activated by the time clotting is completed, especially at relatively high initial sample concentration of protein. For such FVIII, the assay may result in possible underestimation of the protein activity compared to CS assay and in vivo. In any case, it is not possible to determine which of these assays better reflects clinically relevant potency of a particular EHL FVIII variant; thus, the use of any of these assays is to be supported by clinical study. However, these considerations may improve our understanding of the underlying causes of the CS and OC assay discrepancies related to the EHL FVIII molecular structure.

Our hypothesis correlates well with a number of FVIII molecule cleavages by thrombin (FXa) needed to release the bound VWF entity, being from one cleavage for a native-like FVIII heterodimer (FL-FVIII), two cleavages for a single-chain FVIII (rFVIII-SC), and to three cleavages for BIVV001 with the fused D’D3 fragment (an equivalent of the bound VWF), where respective underestimation of FVIII activity by OS assay progressively decreased up to ~60% compared to the CS assay results. Thus, the activation rate of each of these molecules decreases proportionally to the number of its thrombin cleavage sites to release the bound VWF moiety. At the same time, other modifications of FVIII may also contribute to the overall thrombin cleavage rate of the molecule and affect the activity measurement in OC assay, being a factor causing the OC/CS assays’ discrepancy.

Our hypothesis is consistent with data of Hubbard et al. who (i) found that the FVIII activity results decreased upon addition of VWF to samples with FL-rFVIII, pdFVIII, and a BDD-FVIII, and (ii) showed that the observed discrepancy between the used OC and CS assays results might be caused by differences in the activation profiles between these FVIII variants [119].

Notably, the structural differences in FVIII, retained by its activated form and decreasing the rate of FXa generation, would affect both OC and CS assays. In particular, only two EHL FVIII variants, N8-GP, and rFVIII-SC, upon activation, retain high structural similarity to wild-type FVIIIa, whereas other EHL variants still carry PEG or Fc-fusion, etc., that can affect FXa generation rate in both assays.

Our explanation of the OC/CS assay discrepancy is also applicable to the activity measurements of EHL FIX products, used to treat FIX congenital deficiency, also resulting in abnormal bleeding (hemophilia B). Indeed, both FVIII and FX are binding partners within the tenase complex; thus, deficiency in FIX similarly affects the FXa production. There are three current therapeutic EHL FIX variants, which are based on (i) Fc-fragment fusion (eftrenonacog alfa), (ii) pegylation (nonacog beta pegol), and (iii) albumin fusion (albutrepenonacog alfa). For measurement of FIX activity, the respective assay setups differ only by FIX depletion: (i) in plasma used for OC assay, and (ii) in solution used for CS assay step 1, containing FXIa (for FIX activation) and FVIII [87]. Similarly to EHL FVIII, the EHL FIX products are also known to have OC/CS assay results discrepancy, compared to nonmodified FIX comparator (standard). This discrepancy also depends on conditions (reagents, etc.) and the nature of modified FIX [86]. For the latter factor, we similarly propose that the underlying root cause of the assays discrepancy is also a different rate of the modified protein activation (and, possibly, rate of FXa generation by the tenase complex) in OC assay.

### 7.3. Other Research Variants of EHL FVIII

Of interest also are other EHL variants of FVIII described in the literature. FVIII with 237 aa of the B-domain N-terminus (FVIII-237) and truncation of its other parts had a ~1.6-fold extension of the plasma half-life in mice, yet retained affinities to LRP1 and VWF. In VWF-deficient mice, FVIII-237 showed ~10-fold extension of half-life, indicating inhibition of its interactions with endogenous clearance determinants. The 237 aa portion was found to interact with another part of the FVIII molecule, and the authors proposed that the resulting conformation may interfere with interactions with cellular determinants responsible for preconcentration of FVIII, in turn, required for its efficient binding to clearance receptors [120] (as we discussed in Section 4.3).

In another FVIII variant, the B-domain was replaced with a duplex of a nanobody recognizing the D’D3 domains of VWF that resulted in extremely high affinity of the modified FVIII (FVIII-KB013bv) to VWF (K_D_ ~13 pM). Consistently, the plasma half-life of this FVIII variant was ~2-fold prolonged in mice [121].

In our recent study, we investigated the possibility of an FVIII fusion with a single-chain variable fragment (scFv), recognizing FVIII C1-domain with high affinity and inhibiting FVIII binding to LRP1 and VWF, with the idea to affect both pathways of FVIII plasma clearance. However, upon thrombin-mediated disintegration of this scFv, its subunits still remained bound to FVIIIa and interfered with its cofactor function [122]. Lessons learned from this study indicate that affinity of such intramolecular ligand to FVIII should be reasonably lower to allow its dissociation upon thrombin cleavage.

In addition to protein modifications, it is worth asking if the carbohydrate portion of FVIII can be modified to improve its pharmacokinetics. Indeed, this portion is recognized by several clearance receptors that markedly affected FVIII interactions in model systems (Section 4). However, Fay et al. showed that the removal of carbohydrates from FVIII did not affect its biological turnover in rabbits [123]. Consistently, removal of the heavily glycosylated B-domain from FVIII does not affect its plasma-half-life in humans [124]. Thus, it is unlikely that modification of FVIII carbohydrates can result in a clinically meaningful increase of its plasma half-life, though the origin of cells in which rFVIII is expressed affects its glycoprofile [58] and plasma clearance rate in mice [120,125]. In any case, it is still interesting to investigate, in particular, if a hyper sialylation of FVIII prolongs its aging time, thus postponing its interactions with ASGPR and MGL, and extending the plasma half-life similarly to that shown for other proteins [126].

### 7.4. Future Directions

Future studies will evaluate other approaches of FVIII plasma lifetime prolongation to improve the therapeutic efficacy. In particular, better combinations of the favorable molecular modifications may be used on the basis of the codon-optimized FVIII gene to increase protein expression [127].

These principles are likely applicable to improving the gene therapy of hemophilia A, a new approach to treat the disease [128]. Indeed, a transgenic EHL FVIII would require using a smaller amount of the recombinant virus, resulting in a smaller number of the transformed host cells to maintain the target FVIII circulatory level. However, due to limitation of the vector (adeno-associated virus) capsid capacity, only a small increase of the FVIII gene would be possible. In particular, such changes could make the protein (BDD-FVIII, codon-optimized) single-chained or bearing small mutations that decrease its interactions with clearance receptors. Due to this limitation, it is unlikely to achieve protein independence on the VWF clearance pathway requiring fusion of a large protein portion; thus, it is likely that only the FVIII-dependent pathway can be affected.

## Figures and Tables

**Figure 1 ijms-24-08584-f001:**
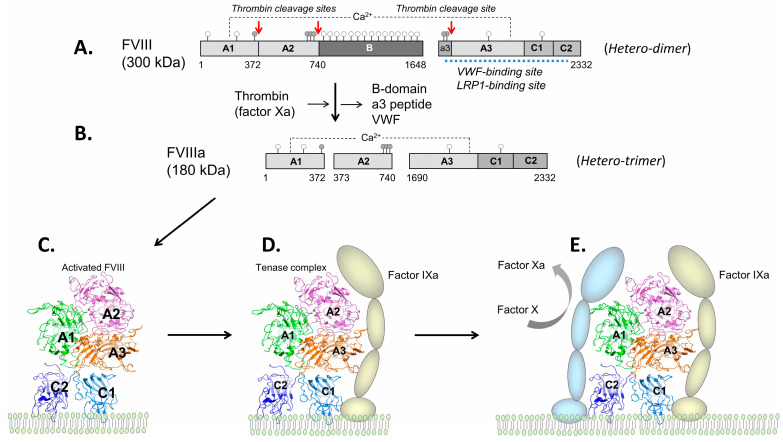
FVIII structure and function. (**A**) FVIII is composed of the heavy chain (HCh, with domain structure A1-A2-B) and light chain (LCh, with domain structure A3-C1-C2) generated during biosynthesis upon the cleavage of the initially synthesized single-chain polypeptide by furin after R1648 [4]. The HCh is heterogenous in length due to variable C-terminal truncations of the B-domain. Thrombin cleavage sites (after R372, R740, and R1689) are marked with red arrows, glycosylation sites are depicted by open circles, sulfated tyrosine residues (Y346, Y718, Y719, Y723, Y1664, and Y1680) are depicted by gray circles, and VWF- and LRP1-binding sites are marked by dotted blue line. (**B**) Upon thrombin (or FXa) cleavage, activated FVIII (FVIIIa) becomes a heterotrimer A1/A2/A3′-C1-C2, while the B-domain, the a3 peptide, and VWF dissociate. This is followed by FVIIIa (*PDB 6MF2*) binding to a platelet membrane (**C**), where it interacts with membrane bound FIXa (protease) forming a complex (tenase) (**D**). In this complex, FVIIIa interacts with membrane-bound FX and facilitates its cleavage (activation) by FIXa (**E**). Generated FXa (protease) participates in further reactions of the blood coagulation pathway.

**Table 1 ijms-24-08584-t001:** Plasma clearance receptors of FVIII and VWF.

Receptor	Recognition Determinants	Expression (Cell Type)	Ligand	Reference
LRP1 ^1^	Protein (specific lysines)	Hepatocytes, macrophages	FVIII, VWF	[14,15]
LDLR ^2^	Protein (specific lysines)	Hepatocytes, macrophages	FVIII	[16]
HSPGs ^3^	Protein (basic residues)	Hepatocytes, macrophages	FVIII	[17]
ASGPR ^4^	Asialic sugars (N-linked)	Hepatocytes	FVIII, VWF	[18,19]
SIGLEC5 ^5^	Sialic sugars (proteins)	Macrophages (etc.)	FVIII, VWF	[20]
CLEC4M ^6^	Mannose sugars	Endothelium	FVIII, VWF	[21,22]
STAB2 ^7^	Glycosaminoglycans	Endothelium	VWF	[23]
SCARA5 ^8^	Glycans	Endothelium, macrophages	VWF	[24]
SCARA1 ^9^	Glycans	Macrophages	VWF	[25]
MGL ^10^	Asialic sugars (O-linked)	Macrophages	FVIII, VWF	[26,27]

^1^ LRP1, low-density lipoprotein receptor-related protein 1; ^2^ LDLR, low-density lipoprotein receptor; ^3^ HSPGs, cell-surface heparan-sulfate proteoglycans; ^4^ ASGPR, asialoglycoprotein receptor; ^5^ SIGLEC5, sialic-acid-binding immuno globulin-like lectin member 5; ^6^ CLEC4M, C-type lectin domain family 4 member M; ^7^ STAB2, stabilin-2; ^8^ SCARA5 (SR-A5), scavenger receptor type A member 5; ^9^ SCARA1 (SR-AI), scavenger receptor type A member 1; ^10^ MGL, macrophage galactose-type lectin.

**Table 2 ijms-24-08584-t002:** Extended plasma half-life rFVIII therapeutic products available in the US.

Molecule Name ^1^	Producing Cells ^2^	Major Modifications ^3^	Available Since ^4^	Reference ^5^
Efmoroctocog alfa ^6^	HEK	BDD, Fc	2014	[88]
Rurioctocog alfa pegol ^6^	CHO	Pegylation ^8^	2015	[89]
Damoctocog alfa pegol ^6^	BHK	BDD, Pegylation ^9^	2018	[90]
Turoctog alfa pegol ^6^	CHO	BDD, Pegylation ^10^	2019	[91]
Efanesoctocog alfa ^7^	HEK	BDD, SCh, Fc, D’D3, XTEN	2023	[92]

^1^ International nonproprietary name of the FVIII variant used as an active ingredient in product. ^2^ HEK, human embryonic kidney cells; CHO, Chinese hamster ovary cells; BHK, baby hamster kidney cells. ^3^ Major modifications: BDD, B-domain deletion; Fc, C-terminal fusion with an antibody Fc fragment; SCh, removal of the furin cleavage site to make the single-chain polypeptide; D’D3, C-terminal fusion with the D’D3 domains of VWF; XTEN, fusion with two XTEN polypeptides (within FVIII and by the D’D3 domains). ^4^ Approval year by U.S. Food and Drug Administration. ^5^ Prescribing information. ^6^ Plasma half-life does not exceed the threshold posed by the plasma half-life of VWF. ^7^ Plasma half-life exceeds the threshold posed by the plasma half-life of VWF. ^8^ Chemical pegylation of FVIII via lysines (random). ^9^ Chemical pegylation of FVIII via the A3 domain C1804 (site-specific). ^10^ Enzymatic pegylation of FVIII via the B-domain linker (site-specific).

## Data Availability

The manuscript does not contain new research data of the author. The previously obtained research data relevant to the manuscript have been published and are referred to.

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
