# Peer review of "Plasma Clearance of Coagulation Factor VIII and Extension of Its Half-Life for the Therapy of Hemophilia A: A Critical Review of the Current State of Research and Practice"

_ijms, 2023, doi:10.3390/ijms24108584_

Round 1
Author Response
Response
We are very thankful to you for taking time to review our manuscript and its high appreciation, very flattering for us. Indeed, we have been working in this field since our co-discovery of the major clearance receptor of FVIII (LRP1) in 1999, including our current regulatory work in U.S. FDA to review applications for FVIII-based drug products, involving EHL those for which we have a special dedication due to our experience in research in FVIII plasma clearance mechanisms.
Regarding your question/consideration about the two FIX fused proteins (Fc-fragment and Albumin fused), we believe that the fusion of both different moieties to FIX results in both similarities and differences in behavior between the two molecules depending on assay conditions in which their biological functions are testedIndeed, the relatively distant fusion site in each on FIX from the location of its activation sites suggests similarity of their activation rate. On the other hand, the difference in structural moieties of both chimeric proteins cannot preclude, and furthermore suggest, difference in their behavior in certain conditions. In particular, such a difference may be related to contributions of the fused portions in interactions with other ligands, especially considering that of the tenase complex members, as both fused portions are not cleaved off upon the molecules’ proteolytic activation.
Indeed, these molecules, albumin-fused FIX (albutrepenonacog alfa (trade name IDELVION)) and Fc-fused FIX (eftrenonacog alfa (trade name ALPROLIX)), show similarity in the activity assays such as (i) underestimation of activity in OC assay when kaolin-based aPTT reagents are used, and (ii) overestimation of activity when ellagic acid-based aPTT reagents are used. At the same time, difference in the molecules’ activities was observed with using other aPTT reagents, and even in a CS assay that showed underestimation of activity for IDELVION, but not for ALPROLIX (Müller et al, Hämostaseologie 2022;42:248–260; ref. [86] in the manuscript). However, we limit this discussion in responding to your concern here and did not include it in the paper to be adherent to its major logical line.

Reviewer 2 Report
Sarafanov presents a review or the current state of therapy involving administration of coagulation Factor VIII and prolongation of its circulatory half-life during hemophilia A therapy.
The introduction sets the stage for his presentation, presenting the rationale for the review and well outlining the contents of the article. Next, the basic biology of FVIII and clinical pathology of a deficiency or absence of activity is also presented. Figure 1 is critical to the reader’s understanding and is an excellent stand-alone piece.
The section concerning not just cell type but also receptor type was new territory for this reader, and the author made this relatively new data understandable.
The following text involving the rationale design of modified FVIII moieties and difficulties assessing factor activity thereafter are also easily understood and presented in an orderly fashion.
Thereafter, the six products that have been assessed so far were displayed with an excellent mini-review of each product.
The remainder of the manuscript integrates the various modified factors with monitoring issues, also illuminating the mechanistic insights provided by these difficulties. Lastly, the section concerning future directions was brief but well supported by the preceding body of this review.
In summary, this review article delivers what it proposed. I have no important criticisms.
Author Response
Response
We are very thankful to you for taking time to review our manuscript and its positive appreciation. Indeed, we have been working in this field since our co-discovery of the major clearance receptor of FVIII (LRP1) in 1999, including our current regulatory work in U.S. FDA to review applications for FVIII-based drug products, involving EHL those for which we have a special dedication due to our experience in research in FVIII plasma clearance mechanisms.
